# Spatial distribution of *Plasmodium vivax* Duffy Binding Protein copy number variation and Duffy genotype, and their association with parasitemia in Ethiopia

Yasin Nasir[1,2ʘ‡], Eshetu Molla[1,2ʘ‡], Getnet Habtamu[1], Solomon Sisay[1], Legesse Alamerie Ejigu[1], Fikregabrail Aberra Kassa[1], Mulugeta Demisse[1], Wakweya Chali[1], Melat Abdo[1], Dawit Hailu Alemayehu[1], Lina Alemayehu[1], Alemayehu Letebo[1,3], Tadele Emiru[1], Jimma Dinsa Deressa[1], Tajudin Abdurhaman Hamza[1,4], Abel Beliyu Tamirat[1,5], Tadesse Misganaw[1,6], Alayu Bogale[1,7], Zufan Yiheyis Abriham[1,8], Sisay Dugassa[2], Migbaru Keffale[1], Fekadu Massebo[3], Hassen Mamo[2], Endalamaw Gadisa[1], Chris Drakeley[9], Alemayehu Godana Birhanu[2], Cristian Koepfli[10], Fitsum G Tadesse[1,9]*

1 Armauer Hansen Research Institute, Addis Ababa, Ethiopia, 2 Addis Ababa University, Addis Ababa, Ethiopia, 3 Arba Minch University, Arba Minch, Ethiopia, 4 University of Werabe, Werabe, Ethiopia, 5 Madda Walabu University Goba Referral Hospital, Bale Goba, Ethiopia, 6 Woldia University, Woldia, Ethiopia, 7 Dilla University, Dilla, Ethiopia, 8 University of Gondar, Gondar, Ethiopia, 9 London School of Hygiene and Tropical Medicine, London, United Kingdom, 10 University of Notre Dame, Notre Dame, Indiana, United States of America

ʘ These authors contributed equally to this work.
‡ These authors are joint senior authors on this work.
* fitsum.girma@ahri.gov.et

## Abstract

### Background

Duffy Binding Protein (PvDBP) binding to the Duffy antigen receptor for chemokine (DARC) is essential for *Plasmodium vivax* invasion of human reticulocytes. PvDBP copy number variation (CNV) might increase parasite invasion and thus parasitemia. We examined the spatial distribution of PvDBP CNVs and DARC genotypes and their association with parasitemia in *P. vivax* endemic settings in Ethiopia.

### Methodology/Principal findings

*P. vivax* isolates (n = 435) collected from five *P. vivax* endemic settings in Ethiopia were genotyped by amplifying the GATA1 transcription factor-binding site of the Duffy blood group and the CNV of PvDBP was quantified. Parasitemia was determined using 18S-based qPCR. The majority of participants were Duffy positive (96.8%, 421/435). Of the few Duffy negative individuals, most (n = 8) were detected from one site (Gondar). Multiple copies of PvDBP were detected in 83% (363/435) isolates with significant differences between sites (range 60%-94%). Both heterozygous (*p = 0.005*) and homozygous (*p = 0.006*) patients were more likely to have been infected by parasites with multiple PvDBP copies than Duffy negatives. Parasitemia was higher among the Duffy positives (median 17,218 parasites/µL; interquartile range [IQR] 2,895–104,489) than Duffy negatives (170;

**Data availability statement:** The data is directly available using (https://doi.org/10.5061/dryad.7pvmcvf40) and the R codes are available using the sample GitHub link provided before (https://github.com/solsisay/Plasmodium-vivax-Duffy-binding-protein-copy-number-variation-and-Duffy-genotype).

**Funding:** FGT was supported by the Bill and Melinda Gates Foundation (ACHIDES; INV-005898, EMAGEN; INV-035257, and HAMMS; INV-048214) and the Wellcome Trust Early Career Award (UNS141457). EM and YN were supported by the Armauer Hansen Research Institute through its core funding from NORAD and SIDA. FGT received a salary from Bill and Melinda Gates Foundation and Wellcome Trust. The funders had no role in the study design, data collection and analysis, decision to publish, or preparation of the manuscript.

**Competing interests:** The authors have declared that no competing interests exist.

78–24,132, $p = 0.004$) as well as in infections with 2 to 3 PvDBP copies (20,468; 3,649–110,632, $p = 0.001$) and more than 3 PvDBP copies (17,139; 2,831–95,946, $p = 0.004$) than single copy (5,673; 249–76,605).

## Conclusions/Significance

A high proportion of *P. vivax* infection was observed in Duffy positives in this study, yet few Duffy negatives were found infected with *P. vivax*. The significant prevalence of multi-copy PvDBP observed among Ethiopian *P. vivax* isolates explains the high prevalence and parasitemia observed in clinical cases. This suggests that vivax malaria is a public health concern in the country where the Duffy positive population predominates. Investigating the relative contribution to the maintenance of the infectious reservoir of infections with different genotyping backgrounds (both host and parasite) might be required.

### Author summary

*P. vivax* establishes infection in humans using its DBP which binds to the DARC receptor on host erythrocytes. A single point mutation in the DARC promoter region results in Duffy negative phenotype. This phenotype results in lower infection success of *P. vivax* and can explain the heterogeneity in the global geographic distribution of *P. vivax*. Here, we investigated the association between Duffy genotypes, CNV of PvDBP, and level of parasitemia using 435 *P. vivax* isolates from five different endemicities in Ethiopia. The results revealed that the majority of *P. vivax* infections do occur in Duffy positive individuals. This explains the high prevalence of *P. vivax* in Ethiopia. The high prevalence of multi-copy PvDBP along with the high rate of Duffy positivity explains the transmission success of *P. vivax* in the country, where the majority of the population are Duffy positive. The lower parasitemia levels among the Duffy-negatives, despite small samples, may signify an "undetected silent reservoir".

## Introduction

Malaria remains a significant public health problem globally with Africa bearing the major burden in terms of the number of cases and deaths, mainly attributable to *Plasmodium falciparum* [1]. Outside of Africa, *P. vivax* is the predominant human *Plasmodium* species responsible for more than half of malaria-associated cases, particularly in Afghanistan, India, and Pakistan [1]. Malaria in Africa has historically been linked to *P. falciparum* although recent reports demonstrated evidence of *P. vivax* transmission in different countries across the continent [2–4]. The infrequent presence of vivax malaria in western and central Africa is likely attributed to high Duffy-negativity among these populations (88–100%) [5]. However, reports confirmed that *P. vivax* is seen in Africa in areas where Duffy positive and negative individuals live together and in areas where Duffy negative population is predominant [6]. Uniquely, countries in the Horn of Africa such as Ethiopia [7], Eritrea [8], Djibouti [9], Somalia [10], and Sudan [11], are the most affected. In Ethiopia, *P. vivax* accounts for approximately 40% of clinical malaria cases [12] with varying innate susceptibility among populations to patent infection with vivax malaria [13,14].

*P. vivax* invasion of human erythrocytes relies on a protein (Duffy Binding Protein, PvDBP) - receptor (Duffy antigen) interaction on the youngest red blood cells, reticulocytes. The Duffy (Fy) antigen is a protein encoded by the Duffy antigen receptor for the chemokines (*darc*) gene on chromosome 1 [15]. DARC is located on both normocytes and reticulocytes; hence this ligand-receptor interaction does not govern selective entry into reticulocytes [16]. *P. vivax* reticulocyte binding protein (PvDBP) interaction with transferrin receptor (CD71) is important for the initial recognition of reticulocytes and explains its strict preference [17,18]. For a long, it was believed that *P. vivax* cannot infect Duffy negative individuals, which is caused by a point mutation in the promoter region of the DARC gene, and this is the reason that *P. vivax* is rare or absent from most of Africa [5,19]. More recent studies confirmed that Duffy-negativity does not confer complete protection against *P. vivax* infections [6,20–24]. The point mutation in the promoter does not necessarily abolish the expression of the Duffy antigen [22,25]. A recent study demonstrated that a subset of Duffy negative erythroblasts transiently express DARC and can thus be invaded by *P. vivax* merozoites [26]. Hence, Duffy negativity is confirmed to not be Duffy null [27,28]. A study in Madagascar showed that *P. vivax* is capable of infecting human erythrocytes without the Duffy antigen [24]. Recently, a similar line of argument emerged to explain the unexpected prevalence of *P. vivax* in Africa.

Amplification of the PvDBP gene copy number has been reported from Ethiopia and elsewhere [29,30]. The parasite's DBP duplication is an important strategy for reticulocyte invasion [30]. PvDBP duplication might allow for binding to an alternative lower affinity receptor in Duffy negative reticulocytes or supports successful infection in Duffy negative patients who may have low level expression of the DARC gene [31,32]. In addition, the multiplication of the PvDBP gene may allow *P. vivax* to evade the host anti-PvDBP immune system [30]. However, the association of DARC genotyping and PvDBP CNVs with *P. vivax* parasitemia is not clearly established. Findings varied from similar parasitemia between heterozygous and homozygous Duffy positive individuals [21,22,33,34] to higher *P. vivax* parasitemia in homozygous than heterozygous individuals [20,35]. In terms of PvDBP CNVs, results varied from a study in Ethiopia that did not find an association with parasitemia [36] to another study in Sudan that showed an association of PvDBP duplications with increases in parasitemia levels [21]. As Ethiopia is a co-endemic setting, the different biological features of *P. vivax* compared with *P. falciparum* such as the hypnozoite stage, evidence of *P. vivax* infections in Duffy negative population in addition to Duffy positives as well as the presence of vectors with wide ranges of feeding and breeding behavior [37] may challenge malaria control and elimination activities.

Ethiopia, with its diverse population of Duffy positive and Duffy negative population [5,38], may provide insights into the epidemiology of *P. vivax.* In this study, the distribution of CNV of PvDBP and DARC genotyping and their association with parasitemias were investigated using *P. vivax* isolates from five different endemic settings in Ethiopia.

## Methods

### Ethical approval

The study protocol was approved by the Armauer Hansen Research Institute and All Africa Leprosy Rehabilitation and Training Center Ethics Review Committee (PO/46/20). Before sample collection, informed written consent and/or assent were obtained from all participants and/or the parents/guardians of the children.

### Study setting, sampling, and sample collection

This study was carried out in five *P. vivax* and *P. falciparum* co-endemic settings in Ethiopia (Adama, Arba Minch Zuria, Batu, Dilla, and Gondar Zuria) from July to October 2021. A total

of 435 symptomatic (n = 374) and asymptomatic (n = 61, only from Dilla Town) participants infected with *P. vivax* mono- (n = 415) and mixed-*P. vivax* and *P. falciparum* (n = 20) infections were included in this study. From each site, 86 to 88 participants were enrolled. Blood samples were collected from symptomatic *P. vivax* patients to investigate the association of *P. vivax* parasitemia (expressed in parasite copies per mL of blood) with host Duffy genotypes and PvDBP copy numbers, with additional asymptomatic community members recruited from one of the study sites (Dilla) in the same period as with clinical cases.

Five-year national malaria surveillance data reported from each of the selected study sites between January 2018 and December 2022, captured by the District Health Information System 2 (DHIS2), were used to analyze the trend and proportions of *P. falciparum* and *P. vivax* diagnosed by rapid diagnostic test (RDT) or microscopy in the five study sites.

Finger prick blood samples (approximately 0.5 mL) collected into EDTA microtainer tubes were used to diagnose malaria using an RDT (SD Bioline Malaria Ag Pf/Pv HRP2/LDH, Standard Diagnostics Inc., South Korea) and microscopic examination of Giemsa-stained blood films, and to prepare dried blood spots (DBS) on Whatman 3 MM filter paper. The DBS samples were allowed to air dry before being kept in zip-locked bags with self-indicating silica gel desiccant beads. The leftover blood in the EDTA tubes stored and shipped at −20 °C was used to extract genomic DNA using the MagMAX magnetic bead-based platform on the Kingfisher Flex robotic extractor (ThermoFisher Scientific) following the manufacturer's protocol [39].

### Plasmodium parasite species confirmation and quantification

Parasite species confirmation and quantification were done in a multiplex qPCR assay that targeted the 18 S rRNA genes for *P. vivax* and *P. falciparum* using primer and probe sequences described earlier [40,41] and TaqMan Fast Advanced Master Mix (Applied Biosystems) on a Bio-Rad CFX96 Real-Time PCR thermocycler (Bio-Rad Laboratories, Inc.). *P. vivax* copy numbers were quantified using serial dilutions ($10^7$ to $10^4$ copies per mL) of recombinant plasmids that contained the amplicon, in duplicate per reaction plate.

### DARC genotyping and copy number variation analysis of PvDBP genes

A qPCR-based TaqMan assay was used to analyze a point mutation in the *darc* gene GATA-1 transcription factor binding region, using previously reported primers and probes [22]. Previously confirmed samples were used as positive controls and molecular grade water as a negative control. The SYBR Green qPCR detection technique was used to determine the PvDBP CNV [30]. The CNV of the PvDBP gene was quantified relative to the single-copy β-tubulin gene (housekeeping gene). Briefly, PCRs were carried out in a total reaction mixture of 20 μL that contained 10 μL 2 X GoTaq qPCR Master Mix, 0.5 μL of each primer (at a concentration of 200 nM), and 2 μL of DNA extract. To detect the specificity of PCR amplification, a melting curve analysis was conducted between the temperature 65 °C and 95 °C with 0.5 °C increments. The PvDBP gene copy numbers were estimated by using synthetic gene (*P. vivax* β-tubulin and PvDBP) combined in varying ratios ranging from 1:1 to 1:6 (one-to-one copy of β-tubulin and one-to-six copies of PvDBP) as described before [42]. The difference in cycle threshold (ct) values between the first mix and 2 to 6 were equivalent to 1, 1.6, 2, 2.3, 2.8 respectively, (i.e., log2^x where x is the ratio of the mix). To determine PvDBP copy number, the equation that was previously employed for PvDBP copy number estimation was utilized as follows: N = 2^-ΔΔCt, where ΔΔCt = (Ct pvβ-t cal- Ct PvDBP cal) - (Ctpvβ-t - CtPvDBP). The Ctpvβ-t and CtPvDBP are threshold cycle values for the *P. vivax* β-tubulin and PvDBP genes respectively, whereas Ctcal is an average difference between Ctpvβ-tubulin and CtPvDBP obtained for the positive control. Then, PvDBP gene copy numbers were categorized

into single copy (isolates with PvDBP estimates of ≤ 0.30), two to three copies (isolates with PvDBP estimates between 0.30 and ≤ 1.01), and more than three copies (PvDBP estimates > 1.01) [36].

## Statistical analysis

Statistical analyses were performed using STATA (version 17.0, Stata Corp., TX, USA) and R (version 4.3.3, the R Foundation for Statistical Computing). The R codes are available using the sample GitHub link (https://github.com/solsisay/Plasmodium-vivax-Duffy-binding-protein-copy-number-variation-and-Duffy-genotype) and the data is directly available using https://doi.org/10.5061/dryad.7pvmcvf40 [43,44]. The sample size was calculated using a single population proportion formula assuming duplication of PvDBP to be 56% based on previous studies conducted in Ethiopia [31], 95% CI, 5% marginal error, and 15% non-response rate. Nonparametric tests were used for pairwise comparisons between variables. The Wilcoxon rank sum test was used to compare parasitemia between symptomatic and asymptomatic cases. The Kruskal-Wallis test was used to compare median parasitemia differences among age groups, Duffy genotypes, and PvDBP gene copy numbers. The Dunnett test was employed to test differences between the median parasitemias with variables such as age group, Duffy status, and PvDBP gene copy numbers. Continuous variables were presented as median and interquartile range (IQR). Ordinal logistic regression model was utilized to assess factors associated with PvDBP copy number. A scatter plot was used to determine the linearity of predictors and the target variable (S1–S3 Figs). A generalized additive model (GAM) with gamma link function that accounted for non-linearity was used to examine the association of parasitemia with DARC genotypes, PvDBP gene copy numbers, and age of the participants.

## Results

### *P. vivax* contributes substantially to the malaria burden in Ethiopia between 2018 and 2022

All the five study sites, with variable altitudes, are co-endemic for the two major species, *P. falciparum* and *P. vivax*. Overall, *P. vivax* contributed to 39% of malaria infections, detected by RDT and/or microscopy, reported from the five sites in the last 5 years. Despite slight variations during the years and notable differences between the study sites, *P. vivax* continues to be an important contributor to the overall malaria burden in Ethiopia between 2020 and 2022. The highest proportion of *P. vivax* infections was reported from Batu (~ 55%) with the lowest from Gondar Zuria (~ 32%) (Fig 1 and S1 Table).

The base map of the Ethiopian administrative boundary shapefile (2021) was obtained from the GADM database (https://gadm.org/). The DEM data were sourced from the USGS Earth Explorer (https://earthexplorer.usgs.gov/) via the SRTM sensor. Both datasets are open-licensed, with GADM free for academic use and USGS in the public domain, ensuring compatibility with the CC BY 4.0 license. The map was created using ArcGIS software.

### Characteristics of the study population

A majority of adult (61.1%, 266/435) males (56.3%, 245/435) who were confirmed to have *P. vivax* mono- (95.4%, 415/435) or mixed-species (*P. falciparum* and *P. vivax*) infections (4.6%, 20/435) by qPCR who were included in this study. Of the clinical cases, 98.7% (369/374) were confirmed to have microscopy-positive infections whilst the rest (n = 5) were only positive by qPCR. Of the asymptomatic community infections recruited in Dilla town, 63.9% (39/61) were negative by microscopy and were positive only by qPCR. qPCR detected median

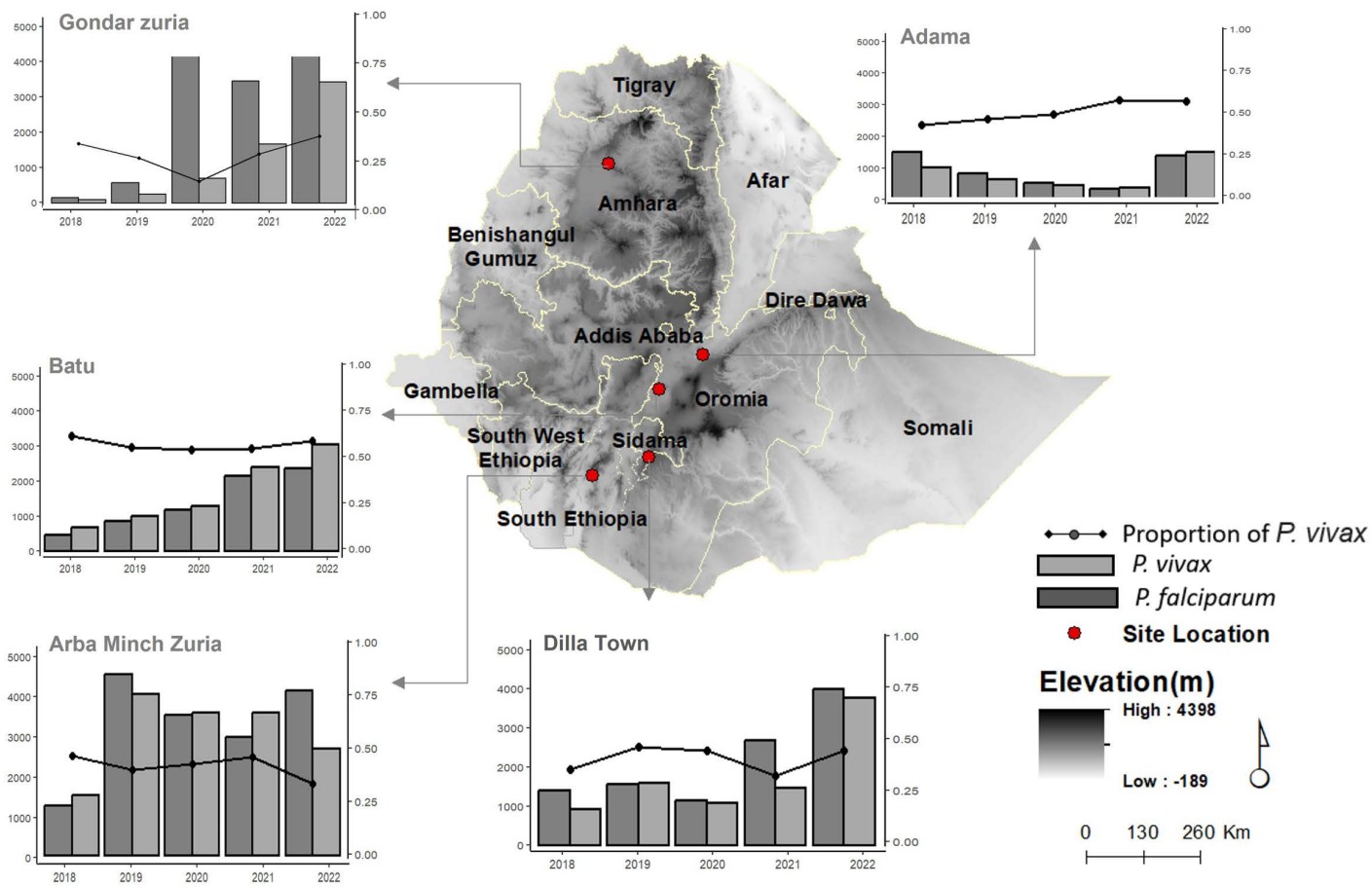

**Fig 1. Malaria trend by Plasmodium species in the study sites in the last five years.** The five-year proportion of *P. vivax* in the study sites is indicated together with the number of reported cases (by species) from the study sites (using DHIS2 data from 2018–2022). The bar plots depict the prevalence of *P. vivax* and *P. falciparum* (left Y-axes) for each year (X-axes) in the study sites. The trend lines (right Y-axes) show the proportion of *P. vivax* by year. The map indicates the study sites with altitude.

parasitemia was different between clinical (17,139; IQR: 2,895–109,895, n = 374) and asymptomatic (8,027; IQR: 1,638–37,204, n = 61, *p* = 0.020) infections (Table 1). The study findings show that the PvDBP copy numbers vary across the study sites (*p* < 0.001, S2 Table).

## DARC genotyping and PvDBP copy numbers vary between study sites

Most of the participants were heterozygous (77.7%, 338/435) or homozygous (19.1%, 83/435) Duffy positive. Only 3.2% (14/435) of the participants were Duffy negative. Duffy genotypes varied between the study sites. All participants in one of the study sites (Batu) were heterozygous Duffy positive whilst the least heterozygosity was detected in Dilla town both in its symptomatic (61.4%, 51/83) and asymptomatic (59.0%, 36/61) communities. Most of the Duffy negative participants were from Gondar Zuria (57.1%, 8/14) (Fig 2A and Table 1).

For PvDBP CNV, 83.4% (363/435) infections had multiple copies; two to three (50.3%, 219/435) or more than three copies (33.1%, 144/435), while only 16.6% (72/435) infections had a single copy. Differences were observed between sites; a higher number of multiple copies were detected in Arba Minch Zuria with two to three 46.6% (41/88) and more than three copies 47.7% (42/88) (Fig 2B and Table 1). The least PvDBP multiple copies were detected in Dilla town: both among the community members (59.0%, 36/61) and the clinical cases (64.0%,

**Table 1. Characteristics of the study participants.**

| Characteristics | Study site | | | | | |
| --- | --- | --- | --- | --- | --- | --- |
| | **Adama** | **Arba Minch** | **Batu** | **Dilla town** | **Gondar Zuria** | **Total** |
| Male sex, % (n/N) | 60.9 (53/87) | 70.5 (62/88) | 50.0 (44/88) | 48.8 (42/86) | 51.2 (44/86) | 56.3 (245/435) |
| Age in years, median (IQR) | 22 (15–31) | 15 (10–20) | 22 (12–27) | 19 (8–34) | 14 (7–20) | 18 (10–28) |
| Species by microscopy, % (n/N) | | | | | | |
| *P. vivax* | 100.0 (87/87) | 93.2 (82/88) | 100.0 (88/88) | 48.8 (42/86) | 100.0 (86/86) | 88.5 (385/435) |
| Mixed | 0.0 (0/87) | 6.8 (6/88) | 0.0 (0/88) | 0.0 (0/86) | 0.0 (0/86) | 1.4 (6/435) |
| Negative | 0.0 (0/87) | 0.0 (0/88) | 0.0 (0/88) | 51.2 (44/86) | 0.0 (0/86) | 10.1 (44/435) |
| Species by qPCR, % (n/N) | | | | | | |
| *P. vivax* | 100.0 (87/87) | 95.5 (84/88) | 100.0 (88/88) | 81.4 (70/86) | 100.0 (86/86) | 95.4 (415/435) |
| Mixed | 0.0 (0/87) | 4.5 (4/88) | 0.0 (0/88) | 18.6 (16/86) | 0.0 (0/86) | 4.6 (20/435) |
| *P. vivax* parasitemia, median (IQR) | | | | | | |
| Symptomatic | 5,683 (1,805–12,765) | 6,257 (1,121–26,389) | 125,632 (91,738–155,631) | 98,833 (77,375– 330,053) | 17,464 (966–98,926) | 17,139 (2,895–109,895) |
| Asymptomatic | NA | NA | NA | 8,027 (1,638– 37,204) | NA | 8,027 (1,638– 37,204) |
| DARC genotype, % (n/N) | | | | | | |
| Positive | | | | | | |
| Heterozygous | 78.2 (68/87) | 68.2 (60/88) | 100.0 (88/88) | 59.3 (51/86) | 82.6 (71/86) | 77.7 (338/435) |
| Homozygous | 20.7 (18/87) | 29.6 (26/88) | 0.0 (0/88) | 37.2 (32/86) | 8.1 (7/86) | 19.1 (83/435) |
| Negative | 1.2 (1/87) | 2.3 (2/88) | 0.0 (0/88) | 3.5 (3/86) | 9.3 (8/86) | 3.2 (14/435) |
| PvDBP copy number, % (n/N) | | | | | | |
| Single copy | 11.5 (10/87) | 5.7 (5/88) | 12.5 (11/88) | 39.5 (34/86) | 14.0 (12/86) | 16.6 (72/435) |
| 2–3 copies | 42.5 (37/87) | 46.6 (41/88) | 58.0 (51/88) | 57.0 (49/86) | 47.7 (41/86) | 50.3 (219/435) |
| > 3 copies | 46.0 (40/87) | 47.7 (42/88) | 29.6 (26/88) | 3.5 (3/86) | 38.4 (33/86) | 33.1 (144/435) |

16/25). Overall, PvDBP copy numbers significantly varied between clinical cases and asymptomatic community samples ($p < 0.001$): multiple copies were detected in 87.4% (327/374) clinical infections versus 59.0% (36/61) asymptomatic community members.

The frequency of multiple PvDBP copies was higher among Duffy positives (84.3%, 355/421) than Duffy negatives (57.1%, 8/14, $p = 0.034$) (S2 Table). Both heterozygous (77.7%, 338/435, $p = 0.005$) and homozygous (19.1%, 83/435, $p = 0.006$) patients were more likely to have been infected by parasites with multiple PvDBP copies than Duffy negatives (Fig 2C and S2 Table).

The base map of the Ethiopian administrative boundary shapefile (2021) was obtained from the GADM database (https://gadm.org/), which is freely available for academic and research purposes.

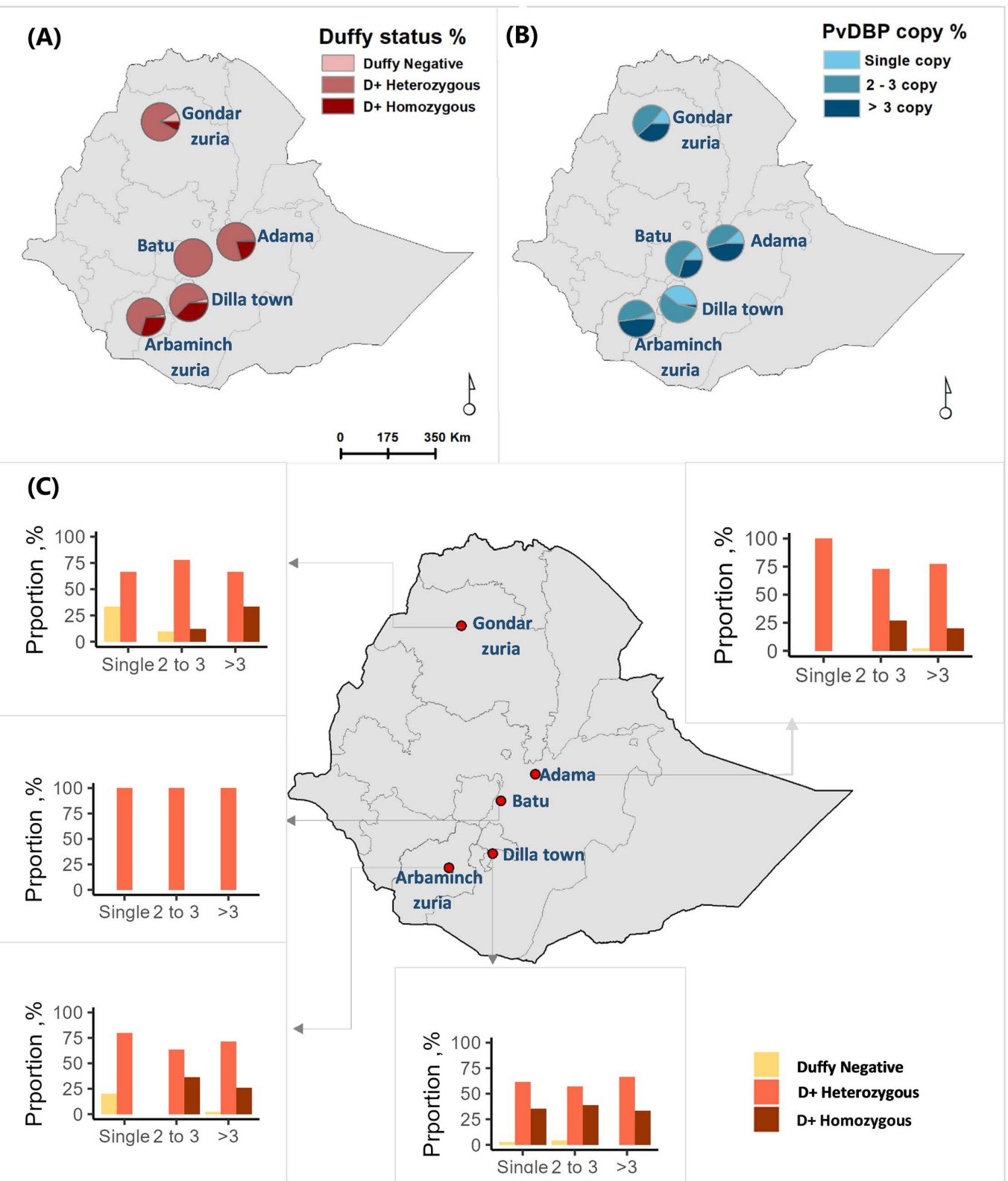

**Fig 2. Geographical distributions of the Duffy genotype and PvDBP copy numbers among *P. vivax* infected participants, Ethiopia.** The pie charts at each site represent the proportion of Duffy genotypes (a), PvDBP copy numbers (b), and bar graphs illustrating the relationship between Duffy genotypes and CNV of PvDBP (c). The Y-axis in the figure panel (c) implies the proportion of individuals with Duffy blood group for each CNV.

## _P. vivax_ parasitemia is associated with Duffy genotype and PvDBP gene copy number

Using a generalized additive model, parasitemia was found associated with the ages of participants, DARC genotyping status, and PvDBP copy number. Parasitemia increases by 0.9 in adults (Density ratio [DR] = 0.9, 95% CI: 0.78–0.95) and children between the age of 5 and 15 (DR = 0.9, 95% CI: 0.80–0.97) as compared to under-fives (Fig 3B and Table 2). Parasitemia was strongly linked with Duffy genotyping status: higher among the Duffy positives (median 17,218 parasites/μL; interquartile range [IQR] 2,895–104,489, n = 421) than Duffy negatives (170; IQR: 78–24,132, n = 14, _p_ = 0.004). Specifically, individuals both with heterozygous (DR = 1.3, 95% CI: 1.07–1.25, n = 338) and homozygous (DR = 1.2, 95% CI: 1.17–1.55, n = 83) Duffy blood groups had increased parasitemia compared to those with the Duffy negative blood group (Table 2). Among the Duffy positive genotypes, median parasitemia was high in heterozygous Duffy blood groups (21,176; IQR: 3,275–111,375) compared to the homozygous Duffy blood group (8,078; IQR: 1,445–36,192, _p_ = 0.009) (Fig 3C). The median parasitemia was also higher in infections with 2 to 3 PvDBP copies (20,468; IQR 3,649–110,632; n = 219, _p_ = 0.001)

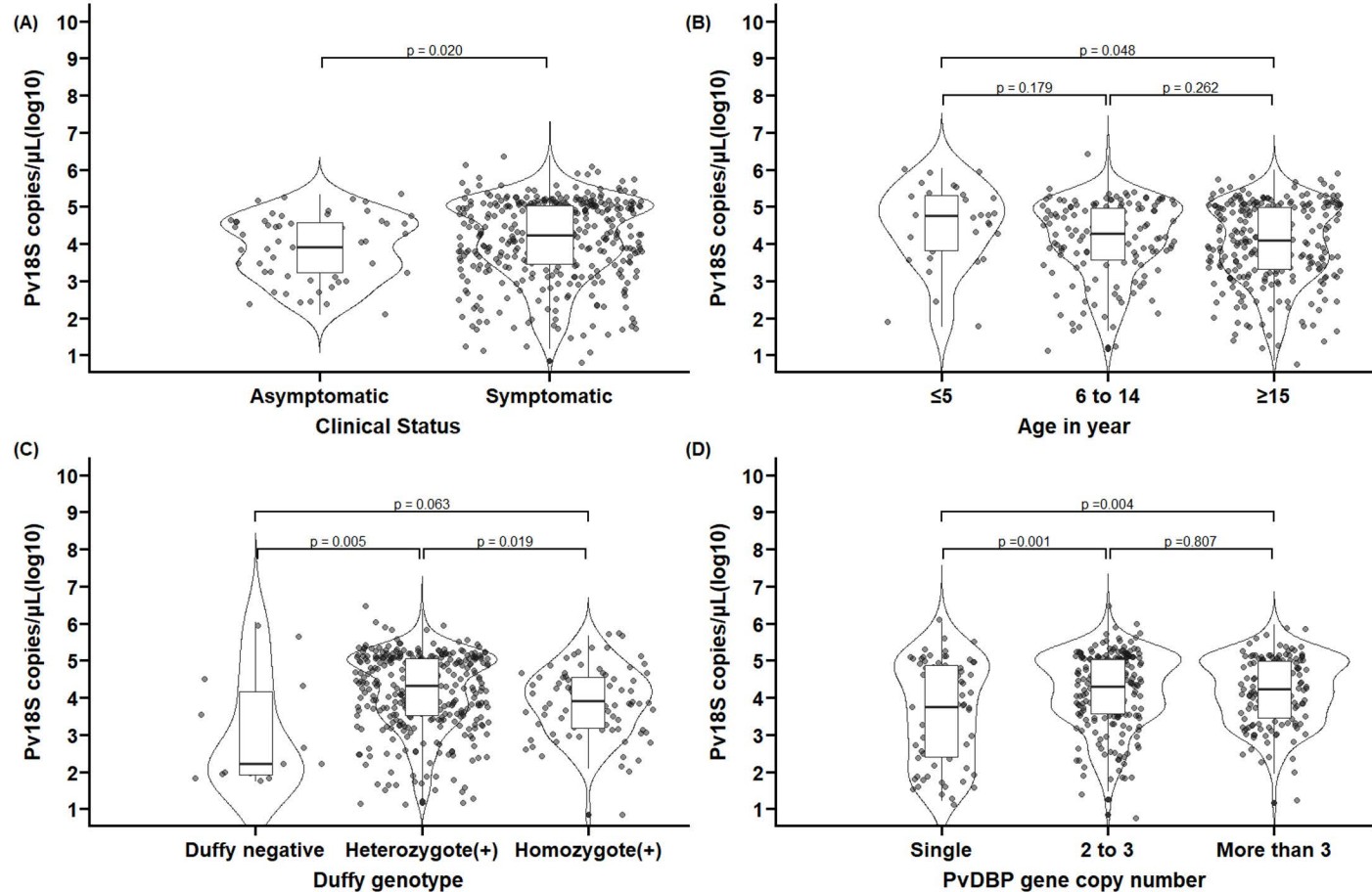

**Fig 3. Comparisons of parasitemia with clinical status, age, Duffy genotypes, PvDBP gene copy numbers of samples based on qPCR assays.** The distribution of parasitemia is shown for symptomatic and asymptomatic individuals (A), age groups (B), Duffy genotypes (C), and PvDBP copy numbers (D) and comparison between groups using the Wilcoxon rank sum test (A) and Kruskal test (B to D). _P_-values were obtained by Dunnett tests, with _P. vivax_ parasitemia (in Log10 Pv18S copies per microliter) as the outcome and age groups, Duffy status, or PvDBP copy numbers as predictors.

**Table 2. *P. vivax* parasitemia association with age group, Duffy genotype, and PvDBP gene copy number.**

| Variables | % (n/N) | DR (95% CI) | *P*-value |
|---|---|---|---|
| Age in years | | | |
| ≤ 5 years (ref.) | 7.4 (32/435) | | |
| 6–14 years | 31.5 (137/435) | 0.9 (0.80–0.97) | 0.015 |
| ≥ 15 years | 61.1 (266/435) | 0.9 (0.78–0.95) | 0.003 |
| Duffy blood group | | | |
| Duffy negative (ref.) | 3.2 (14/435) | | |
| Heterozygous | 77.7 (338/435) | 1.3 (1.07–1.25) | < 0.001 |
| Homozygous | 19.1 (83/435) | 1.2 (1.17–1.55) | 0.003 |
| PvDBP copy number | | | |
| Single (ref.) | 16.6 (72/435) | | |
| 2 to 3 | 50.3 (219/435) | 1.1 (1.07–1.24) | < 0.001 |
| More than 3 | 33.1 (144/435) | 1.2 (1.07–1.25) | < 0.001 |

*A density ratio (DR) with a 95% confidence interval (CI) is given in parentheses.*

and 4 or more PvDBP copies (17,139; IQR 2,831–95,946; n = 144, *p* = 0.004) compared to single copies (5,673; IQR 249–76,605; n = 72) (Fig 3D). Infections with PvDBP gene copy numbers of two to three carried 1.1 times higher parasitemia (DR = 1.1, 95% CI: 1.07–1.24) compared to those with a single copy. Likewise, samples with PvDBP copy numbers of more than three had 1.2 times higher parasitemia (DR = 1.2, 95% CI: 1.07–1.25) compared to those with a single copy (16.6%, 72/435) (Table 2).

## Discussion

In five settings in Ethiopia that are sympatric for the two species, *P. falciparum* and *P. vivax,* the latter contributed to an overall 39% of malaria infections reported in the last five years. We determined the spatial heterogeneity of DARC genotyping and PvDBP CNV and their association with parasitemia in these sites. Most of the *P. vivax* infected individuals in this study were Duffy positive whilst only fourteen Duffy negative individuals were found infected with *P. vivax.* Most of the parasites had multiple copies of the PvDBP gene (83% isolates) with substantial differences between sites, ranging from 94.3% in Arba Minch to only 60.5% in Dilla. Parasites with multiple PvDBP copies were detected more commonly (87.4%) in clinical infections than asymptomatic infections (59.0%) as well as in Dufy positive heterozygous (84.0%) and homozygous patients (85.5%) than Duffy negative patients (61.5%). Our investigation confirmed that parasitemia is associated with age and DARC genotyping status of patients and the number of PvDBP gene copies of the parasites in Ethiopia.

Malaria has been increasing in the last five years in Ethiopia, especially starting from 2020. This may be attributed to service interruptions attributed to the COVID-19 pandemic [45], the emergence of drug and diagnostic-resistant parasites [39,46–48], mosquito resistance to insecticides [1,45], the expansion of the invasive vector *Anopheles stephensi* [39,49–53], and climate change [1,54]. The contribution of *P. vivax* has remained substantial in Ethiopia [55]. The overall burden of malaria linked with *P. vivax* and its trend observed in the present study is in line with the national figure [ 7,12,55].

This study confirms the low but appreciable presence of the *P. vivax* parasite reservoir in Duffy negative populations. Duffy positive genotypes predominate among populations susceptible to *P. vivax* infections [56]. *P. vivax* endemicity is associated with the Duffy gene expression in the population in which this parasite is more endemic in areas where a high

population of Duffy-positives exists [57,58]. Our result was corroborated by a prior report that found 86.9% of heterozygous *P. vivax* patients, followed by 11.7% homozygous for Duffy positive and 1.4% negative. This supports the hypothesis that homozygous individuals are less susceptible to *P. vivax* infection [59]. A recent systematic review also revealed a varied relationship between the Duffy genotype/phenotype and *P. vivax* infection ranging from no evidence of malaria cases in the Duffy negative genotypes to a higher prevalence (99%) of *P. vivax* infections in these groups [60]. Another systematic review also confirmed a low prevalence of *P. vivax* infections among Duffy-negative individuals in East Africa [61]. However, there is limited global evidence to date to substantiate this phenomenon [36]. Our study was limited to addressing the relationship between allelic types and the risk of malaria.

Discovery of the molecular mechanisms of successful infection in Duffy negative and Duffy positive individuals made it clear that Duffy negative individuals are not completely resistant to *P. vivax* infections, but suggest a role in resistance to clinical disease [62]. The high proportion of Duffy positives may serve as a potential source of *P. vivax* infection for Duffy negatives [24]. The high frequency of this phenotype in our population coupled with *P. vivax* endemicity demands tailored interventions targeting *P. vivax*. Yet, it is vital to examine the transmission pathways and the genetic diversity of *P. vivax* among Duffy positive and Duffy negative populations.

The high proportion of *P. vivax* infections with multiple PvDBP copies in this study supported by previous studies, confirmed that multi-copy PvDBP infections are common in Ethiopia [31,36,63]. Gene duplication can generate new gene functions or alter gene expression pathways [64]. We have observed differences between the study sites which might be linked with the parasites' genetic diversity. The highest PvDBP CNV was reported from Arba Minch, where a recent study demonstrated the highest haplotype and nucleotide diversity of *P. vivax* [65]. A higher proportion of multiple PvDBP copy parasites was observed in symptomatic patients compared to the asymptomatic participants, supported by a previous report [36]. As it is well known the parasite uses this ligand-receptor interaction to establish infection in RBCs and limited or enhanced availability or binding of the two pairs might play an important role in determining the parasitemia level. In our previous study including a community cohort where we followed asymptomatic infected participants over a year, some infections progressed to clinical high density infections whilst others remained around the limit of detection (low density infections) throughout the study [66,67]. The inclusion of asymptomatic community members from all study sites could have provided important insights. Our study was limited to include asymptomatic individuals only from one of the study sites. Future investigation with appropriate study design that includes the asymptomatic community might support generating a definitive conclusion.

In our study, we observed higher PvDBP copies in patients with Duffy positive genotypes compared with Duffy negatives although there were very few Duffy negative observations. Similar PvDBP expansion were detected at lower frequencies in Cambodia, Brazil, and India where a small proportion of Duffy negative individuals live [31]. Debatably, PvDBP copy numbers may not be selected in response to the Duffy-negativity barrier [42]. No specific PvDBP sequence polymorphisms that are associated with Duffy negative erythrocyte invasion by the parasites were reported [68]. On the contrary, higher CNVs of PvDBP1 were observed in Duffy negative than positive [32]. Further studies may elucidate the significance of PvDBP CNV in Duffy positive and Duffy negative erythrocyte invasion mechanisms.

Duffy positive *P. vivax*-infected participants had high parasitemia compared with Duffy negatives in this study, in line with other reports from Ethiopia [20,22,33] and elsewhere [21,69]; whilst another report from western Ethiopia reported conflicting results [70]. The high parasitemia observed in the under-five children which might be linked with premature immunity at

an early age suggests an important role they might play in sustaining the transmission reservoir [71]. A significant number of *P. vivax* infections with high parasitemia levels could be a challenge for malaria elimination programs targeting *P. vivax*. On the other hand, the observed low parasite load in Duffy negative individuals might suggest low invasion competence of *P. vivax* [27,33], despite small Duffy negative samples. Parasite invasion of reticulocytes is determined to be reduced in the absence of the Duffy antigen expression [72]. Our data noted that all of the infections detected in Duffy negative individuals were not missed by microscopy albeit low density. Molecular methods might provide better estimates of the silent *P. vivax* parasites reservoir that could remain undetected by routine diagnostic tests (microscopy and RDT). *P. vivax* could survive in Duffy negative individuals without eliciting any symptoms, but the importance of these infections to the infectious reservoir needs further investigation [73]. Based on the low parasite densities detected in this study, Duffy negative vivax patients may be less infectious to mosquitoes but their relative contribution remains to be quantified [22,33].

Within the Duffy positive population, we observed higher parasitemia in heterozygous than homozygous individuals contrasting with other studies [20,22,33], which reported a comparable range of parasitemia between the two groups. This deviation could be due to study site-level variations of parasitemia and Duffy genotype in our study. For instance, in Batu, where all of the patients were heterozygous, the maximum median parasitemia was detected. Contrary to the above studies, a previous study in Ethiopia reported that gametocytemia levels in heterozygous individuals were found to be relatively higher than in homozygous groups [74]. Further investigations are required if these are biologically relevant. Taking into account previous studies that did not find an association between parasitemia and PvDBP copy number [31,36], several lines of arguments might explain the observation in this study. Considering that most infections had multiple copies of PvDBP, the interaction between DARC and PvDBP can be enhanced and that might explain the higher parasitemia we observed in the heterozygous population. Until very recently, the interaction between DARC and PvDBP was solely hypothesized to be behind an invasion of reticulocytes. No evidence exists to date that polymorphisms in the PvRBP2b and CD71 play any role in the successful invasion of reticulocytes and subsequently parasitemia.

Despite the majority of the study participants being Duffy positive, infections have been detected in Duffy negative individuals as well; these were low density in general. The high prevalence of multi-copy PvDBP observed among Ethiopian *P. vivax* isolates, combined with the high rate of Duffy positivity, that were directly related to the observed parasitemia levels corroborates the fact that the parasite evolved to maximize transmission success. These could challenge the malaria elimination efforts of the country. Considering that gametocyte production mirrors asexual parasitemia in *P. vivax*, high density infections are likely to produce more gametocytes and contribute substantially to the maintenance of transmission in endemic settings. In addition, *P. vivax* could survive in Duffy negative individuals without eliciting any symptoms, but the importance of these infections to the infectious reservoir needs further investigation. In general, the relative public health importance of *P. vivax* infections in Duffy positive and Duffy negative individuals, the impact of PvDBP CNVs, and their relative contribution to the maintenance of the infectious reservoir needs to be investigated in detail in natural infections.

## Supporting information

**S1 Fig. A scatter plot of parasitemia versus age with an overlay of a non-linear LOWESS line (red line).** The plot represents the relationship between the variables visually, with the non-linear relationship being represented by the line.
(TIF)

**S2 Fig. Diagnostic plots for assessing linearity assumptions and fitting models (lm model).** These plots can provide insights into the relationship between the parasitemia and the predictor variables (Age, PvDBP copy number, Duffy blood group). **Residuals vs. Fitted:** This plot shows the differences between observed and predicted values (on the y-axis and the fitted values on the x-axis). This helps identify patterns and non-linearities in the relationship between the response and predictors. A good fit is indicated by residuals that are randomly distributed around zero. **Normal Q-Q (Q-Q Residuals) Plot:** This graph compares the quantiles of standardized residuals to those of a normal distribution. It helps determine whether the residuals have a normal distribution. If the points roughly follow a straight line, the assumption of normality is probably met. **Scale-Location Plot:** This plot assists in checking for constant variance (homoscedasticity) of residuals. Ideally, the points should be randomly scattered around a horizontal line. **Residuals vs. Leverage:** This plot helps to identify influential observations that have a large impact on the model fit. Observations with high leverage and large residuals may have a strong influence on the estimated coefficients.
(TIF)

**S3 Fig. Diagnostic plots for assessing fitting models (GAM model).** Observed Vs Predicted: This plot shows the observed parasitemia against the fitted values from the GAM model. The y-axis represents the observed response, and the x-axis represents the fitted values. It helps assess the overall fit of the model. Ideally, the points should fall along a straight line, indicating that the model captures the relationship between the predictors and the response variable. Any systematic deviations from the line could suggest issues with the model's ability to capture the underlying patterns in the data.
(TIF)

**S1 Table. The trend of confirmed *P. vivax* and *P. falciparum* cases in each study site between 2018 and 2022.**
(XLSX)

**S2 Table. PvDBP duplication referring to age category, sex, infection type, Duffy blood group, and site Ethiopia (N = 435).**
(XLSX)

**S3 Table. Result of the ordinal logistic regression model for significantly associated variables with PvDBP copy number, Ethiopia (N = 435).**
(XLSX)

## Acknowledgments

We would like to acknowledge the study participants, the field research team, the community facilitators, and regional and district health officers for their support during sample collection and transportation.

## Author contributions

**Conceptualization:** Yasin Nasir, Eshetu Molla, Cristian Koepfli, Fitsum G Tadesse.

**Data curation:** Eshetu Molla, Getnet Habtamu, Solomon Sisay, Legesse Alamerie Ejigu, Fikregabrail Aberra Kassa, Mulugeta Demisse.

**Formal analysis:** Eshetu Molla, Solomon Sisay, Legesse Alamerie Ejigu, Fikregabrail Aberra Kassa, Mulugeta Demisse.

**Funding acquisition:** Cristian Koepfli, Fitsum G Tadesse.

**Investigation:** Yasin Nasir, Eshetu Molla, Getnet Habtamu, Wakweya Chali, Melat Abdo, Dawit Hailu Alemayehu, Lina Alemayehu, Alemayehu Letebo, Tadele Emiru, Jimma Dinsa Deressa, Tajudin Abdurhaman Hamza, Abel Beliyu Tamirat, Tadesse Misganaw, Alayu Bogale, Zufan Yiheyis Abriham, Migbaru Keffale, Fitsum G Tadesse.

**Methodology:** Yasin Nasir, Eshetu Molla, Solomon Sisay, Fitsum G Tadesse.

**Project administration:** Migbaru Keffale, Fitsum G Tadesse.

**Resources:** Melat Abdo, Migbaru Keffale, Fitsum G Tadesse.

**Software:** Fikregabrail Aberra Kassa.

**Supervision:** Migbaru Keffale, Hassen Mamo, Endalamaw Gadisa, Chris Drakeley, Alemayehu Godana Birhanu, Fitsum G Tadesse.

**Validation:** Fitsum G Tadesse.

**Visualization:** Eshetu Molla, Solomon Sisay, Legesse Alamerie Ejigu, Mulugeta Demisse, Fitsum G Tadesse.

**Writing – original draft:** Yasin Nasir, Eshetu Molla, Solomon Sisay, Mulugeta Demisse, Fitsum G Tadesse.

**Writing – review & editing:** Yasin Nasir, Eshetu Molla, Solomon Sisay, Legesse Alamerie Ejigu, Mulugeta Demisse, Tadele Emiru, Sisay Dugassa, Fekadu Massebo, Hassen Mamo, Endalamaw Gadisa, Alemayehu Godana Birhanu, Cristian Koepfli, Fitsum G Tadesse.

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
