## [Decision Letter · Decision Letter 0]

22 Oct 2024

Dear Dr Tadesse,

Thank you very much for submitting your manuscript "Spatial distribution of Plasmodium vivax Duffy binding protein copy number variation and Duffy genotype, and their association with parasitemia in Ethiopia" for consideration at PLOS Neglected Tropical Diseases. As with all papers reviewed by the journal, your manuscript was reviewed by members of the editorial board and by several independent reviewers. The reviewers appreciated the attention to an important topic. Based on the reviews, we are likely to accept this manuscript for publication, providing that you modify the manuscript according to the review recommendations. 

Sincerely,

Ananias A. Escalante, PhD

Academic Editor

Abhay Satoskar

Section Editor

Reviewer's Responses to Questions

**Key Review Criteria Required for Acceptance?**

**Methods**

-Are the objectives of the study clearly articulated with a clear testable hypothesis stated?

-Is the study design appropriate to address the stated objectives?

-Is the population clearly described and appropriate for the hypothesis being tested?

-Is the sample size sufficient to ensure adequate power to address the hypothesis being tested?

-Were correct statistical analysis used to support conclusions?

-Are there concerns about ethical or regulatory requirements being met?

Reviewer #1: There are some questions and comments in the methodology, so the authors should address and give clarification as per the attached comments

Reviewer #2: The objectives were clear and the study designed allowed for interrogating this. Each if the populations were described although the numbers were no even and asymptomatic infections were only obtained from one population. As this was a single survey for genotyping, there was no indication of samples size determination to allow for power to determine correlations or association between genotypes and clinical variables. However, I believe the number of samples obtained per location allow for suggestive statistical trends and comparable to similar studies conducted. As mostly malaria infected individuals were genotyped, the DARC frequencies will have to described in context of this sub-populations, although they may be representative of a true population sample.

Why were asymptomatic cases only from a single location. It is not clear if these were from the same period as the clinical cases.

**Results**

-Does the analysis presented match the analysis plan?

-Are the results clearly and completely presented?

-Are the figures (Tables, Images) of sufficient quality for clarity?

Reviewer #1: Yes, but there are some questions and comments in the result and discussion, so the authors should address and give clarification as per the attached comments

Reviewer #2: There were variable dynamics in the distribution of P. falciparum vs P. vivax infections across years from the survey data. Unlike reported on Page 8, there is no significant sign of decreasing proportions of P. vivax except for Arba Minch Zuria. All other sites showered similar proportions for 2018 vs 2022. 

We noted the exceptionally high frequencies of Duffy-negative heterozygotes, although the allelic types (FyA or FyB) were not determined. It was surprising that all samples from Batu were heterozygous. These need further verification as it will require extreme positive selection to maintain this or complete purification of homozygous wild or mutants, which is rare in human population. This should be discussed.

To further expand on the relationship between Duffy genotypes and DBP copy numbers, plot of the distribution of proportions of alleles versus CNVs could be informative and add to the data displayed separately on Fig2 A and Fig 2B.

It was surprising that the data was not in line with the allele dosing model in which higher parasitemia will be expected in the homozygous Duffy positives as against the heterozygous and least in the homozygouz Duffy Null. As most individuals were heterozygous, this comparison was highly skewed. It will be informative to check this only for populations that has all combinations of alleles separately. Adding Batu for example in the bulk statistics will exaggerate the heterozygous numbers.

The figures and tables are clear and mostly sufficient

**Conclusions**

-Are the conclusions supported by the data presented?

-Are the limitations of analysis clearly described?

-Do the authors discuss how these data can be helpful to advance our understanding of the topic under study?

-Is public health relevance addressed?

Reviewer #1: Yes

Reviewer #2: The discussion was slightly more elaborate that might be required but well presented with a large number of references. There was a significant discussion on the epidemiology of malaria which distracts from the main theme of Duffy antigen variants, DBP copy numbers and malaria in Ethiopia.

As there further definition of the Duffy blood group types, the relationship with risk to malaria by the allelic type is not as clear as proposed on page 18. Morover, reference 79 was done in Brazil, where the P. vivax has diverged and the human population has experienced a different malaria history. Moreover, the risk of malaria was allele dependent, with Fyb/Fyo being at risk, while Fya/Fyo had higher DBP antibodies that were protective.

In reference to how infection densities contribute to transmission, it was stated on Page 18 that high densities could lead to high gametocytemia and transmission. However, this is against infections in Duffy negative being a reservoir as they mostly had low level parasitemia. The authors should not overlook this.

Overall, the conclusions are valid and of public health relevance. The limitations need to be better presented and discussed.

**Editorial and Data Presentation Modifications?**

Reviewer #1: (No Response)

Reviewer #2: Minor revision as per comments above

**Summary and General Comments**

Reviewer #1: There are some general comments, so the authors should address and give clarification as per the attached comments

Reviewer #2: This is an important piece of work and adds to the body of knowledge about genetic interactions between parasites and human populations. The data is well presented with only minor modifications and need to present the statistical analysis done when stating that differences were seen. An example will be dofifferences in copy numbers for DBP between sites, while no stats was shown. Another is comparing asymptomatics from Dilla with symptomatics from all sites, while Dilla this could be limited to Dilla only. The rationale for comparing the PvDBP CNV between clinical and asymptomatic community samples is not clear considering that there are 374 clinical samples collected from five sites while you have only 61 asymptomatic samples collected from just one site. Moreover, there is no discussion of the symptomatic and asymptomatic results in the discussion section of the manuscript. All other observations for the attention of the authors are commented above.

PLOS authors have the option to publish the peer review history of their article (what does this mean? ). If published, this will include your full peer review and any attached files.

**Do you want your identity to be public for this peer review?** For information about this choice, including consent withdrawal, please see our Privacy Policy .

Reviewer #1: No

Reviewer #2: Yes: Alfred Amambua-Ngwa

Figure Files:

Data Requirements:

Reproducibility:

References

---

## [Editor Report · Decision Letter 1]

13 Jan 2025

Dear Dr Tadesse,

We are pleased to inform you that your manuscript 'Spatial distribution of Plasmodium vivax Duffy Binding Protein copy number variation and Duffy genotype, and their association with parasitemia in Ethiopia' has been provisionally accepted for publication in PLOS Neglected Tropical Diseases.

Best regards,

Ananias A. Escalante, PhD

Academic Editor

Abhay Satoskar

Section Editor

Shaden Kamhawi

co-Editor-in-Chief

Paul Brindley

co-Editor-in-Chief

---

## [Editor Report · Acceptance letter]

Dear Dr Tadesse,

We are delighted to inform you that your manuscript, "Spatial distribution of *Plasmodium vivax* Duffy Binding Protein copy number variation and Duffy genotype, and their association with parasitemia in Ethiopia," has been formally accepted for publication in PLOS Neglected Tropical Diseases.

Best regards,

Shaden Kamhawi

co-Editor-in-Chief

Paul Brindley

co-Editor-in-Chief
